# Human APOBEC3 Variations and Viral Infection

**DOI:** 10.3390/v13071366

**Published:** 2021-07-14

**Authors:** Shiva Sadeghpour, Saeideh Khodaee, Mostafa Rahnama, Hamzeh Rahimi, Diako Ebrahimi

**Affiliations:** 1Department of Biological Science, University of California Irvine, Irvine, CA 92697, USA; sadeghps@uci.edu; 2Department of Bioinformatics, Institute of Biochemistry and Biophysics, University of Tehran, Tehran 1417614335, Iran; s.khodaee@ut.ac.ir; 3Department of Plant Pathology, University of Kentucky, Lexington, KY 40546, USA; mostafa.rahnama@uky.edu; 4Department of Molecular Medicine, Biotechnology Research Center, Pasteur Institute of Iran, Tehran 1316943551, Iran; rahimi.h1981@gmail.com; 5Texas Biomedical Research Institute, San Antonio, TX 78227, USA

**Keywords:** APOBEC3, Polymorphism, Splice variants, HIV, HBV

## Abstract

Human APOBEC3 (apolipoprotein B mRNA-editing catalytic polypeptide-like 3) enzymes are capable of inhibiting a wide range of endogenous and exogenous viruses using deaminase and deaminase-independent mechanisms. These enzymes are essential components of our innate immune system, as evidenced by (a) their strong positive selection and expansion in primates, (b) the evolution of viral counter-defense mechanisms, such as proteasomal degradation mediated by HIV Vif, and (c) hypermutation and inactivation of a large number of integrated HIV-1 proviruses. Numerous APOBEC3 single nucleotide polymorphisms, haplotypes, and splice variants have been identified in humans. Several of these variants have been reported to be associated with differential antiviral immunity. This review focuses on the current knowledge in the field about these natural variations and their roles in infectious diseases.

## 1. Introduction

Human chromosome 22 codes for seven APOBEC3 enzymes (APOBEC3A/B/C/D/F/G/H), at least four of which (APOBEC3D, APOBEC3F, APOBEC3G, and stable haplotypes of APOBEC3H) have anti-HIV activities (reviewed in [1,2,3,4,5,6,7]). These innate immune enzymes can inhibit HIV-1 replication by inflicting C > U hypermutation in viral genomes and/or by deaminase-independent mechanisms [8,9,10,11,12] (Figure 1).

APOBEC3A, APOBEC3C and APOBEC3H have a single cytosine deaminase (CD) domain. By contrast, APOBEC3B, APOBEC3D, APOBEC3F, and APOBEC3G have two CD domains, of which only the C-terminal CD2 is catalytically active. The N-terminal CD1 domains of double-domain APOBEC3 enzymes are catalytically inactive. This is despite the fact that both of the CD domains have the zinc-coordinating motif. The CD1 domain plays a role in binding RNA and viral packaging of APOBEC3 enzymes. It also facilitates ssDNA binding and is required for antiviral activity. Each of the CD1 and CD2 domains are made of five beta sheets and six alpha helices, surrounding a zinc atom. The CD domains have a consensus Zn-binding motif His-X-Glu-X_23–28_-Pro-Cys-X_2–4_-Cys, where X can be any amino acid. There are three types of CD domains (Z1, Z2, and Z3), which are defined by three specific and conserved amino acid sequences of the catalytic domain [13,14,15,16,17,18].

The main substrate for APOBEC3-induced cytosine to uracil (C > U) deamination is single-stranded DNA (ssDNA). APOBEC3G preferentially mutates cytosine within the CC dinucleotides of HIV-1 cDNA during reverse transcription (the targeted cytosine is underlined). These mutations are manifested as GG > AG changes in the viral genome. By contrast, other APOBEC3 enzymes preferentially target C within TC, which leads to GA > AA changes in the HIV-1 genome [8,9,19,20].

The HIV-1 protein Vif counteracts the antiviral activity of APOBEC3 enzymes. Vif binds to APOBEC3 enzymes and mediates their ubiquitination by the cellular Cullin5 E3-ubiquitin ligase, followed by their proteasomal degradation [4,21,22,23,24]. The counteraction ability of Vif depends on both the type of APOBEC3 enzymes and the variations within Vif. For instance, unlike APOBEC3G, APOBEC3F is partially resistant to neutralization by Vif [25]. Another example is APOBEC3H haplotype II, which is highly sensitive to the Vif variants 39F and 48H, but not the variants 39V and N48 [26,27,28]. These differential APOBEC3 interactions with Vif (and possibly with other HIV-1 proteins, such as RT [11,29,30,31] and protease [32]) can have widely different functional consequences. The inability to neutralize highly mutagenic APOBEC3 enzymes, such as APOBEC3G and APOBEC3H haplotype II, often leads to the generation of inactivated HIV-1 genomes with many C > U (G > A) changes, including stop codons [33,34,35,36]. These hypermutated sequences are almost exclusively found as integrated proviruses that are incapable of generating viable progeny viruses. On the other hand, the targeting of the HIV-1 genome by less mutagenic enzymes, such as APOBEC3D and APOBEC3F, or by partially neutralized APOBEC3G or APOBEC3H haplotype II, can generate sub-lethally mutated viruses. APOBEC3-induced sub-lethal mutagenesis can contribute to HIV-1 diversification [35,36,37,38,39], evolution [37,40,41], immune escape [42,43,44], and drug resistance [42,45,46,47,48,49,50].

The function of APOBEC3 enzymes is not limited to HIV-1 restriction. These enzymes have also been implicated in the inhibition and/or evolution of several other viruses, [51] such as hepatitis B virus (HBV) [52,53,54,55], polyomaviruses (e.g., JC and BK) [56,57], human T-cell leukemia virus-1 (HTLV-1) [58,59], human papillomavirus (HPV) [60,61,62], and herpesviruses [63,64] including the Epstein–Barr virus (EBV) [65,66], herpes simplex virus-1 (HSV-1) [65], and Kaposi’s sarcoma-associated herpesvirus (KSHV) [67]. It is important to note that data on the inhibition and hypermutation of herpesviruses by APOBEC3 enzymes are not consistent [65,66,68,69]. Overall, unlike HIV-1, evidence for the in vivo hypermutation of these viruses is extremely limited. Therefore, deaminase-independent mechanisms are likely to be responsible for the inhibition of these viruses by APOBEC3 enzymes. Very little is known about deaminase-independent mechanisms and their contributions to the inhibition of viruses other than HIV-1. For example, APOBEC3G has been suggested to inhibit HIV-1 reverse transcription by direct interaction with HIV-1 RT [29,30], or by binding to viral RNA/ssDNA and limiting the movement of RT [70,71], or by reducing tRNA annealing to HIV-1 RNA [72,73]. However, it is unclear how these processes can inhibit DNA viruses such as HBV, polyomaviruses, and herpesviruses.

APOBEC3 enzymes have been under positive selection, and have been expanded from one gene in mice to seven genes in primates [74,75,76,77,78]. Additionally, they have diversified at both the genomic and transcriptomic levels in human populations. This is possibly in response to exposure to different pathogens in different parts of the world [32]. Similarly, some of the changes observed in viral genomes (e.g., HIV-1 Vif variantions V39F and N48H) can be due to the adaptation to different APOBEC3 variants in different human populations [26,27,28]. These variations can underlie population-specific virus–host interaction mechanisms. This review article focuses on human APOBEC3 variations that have been investigated for their relevance in infectious diseases.

## 2. APOBEC3 Variations

APOBEC3 enzymes have many coding and noncoding variations. They are also diverse in terms of the transcripts (splice variants) they produce. The majority of variations that have been reported for APOBEC3 enzymes are at the DNA level, particularly single nucleotide polymorphisms (SNPs) [79,80,81,82]. Few studies have also investigated APOBEC3 splicing [32,83,84,85]. A summary of the reported variants with roles in the biology of APOBEC3 enzymes is listed in Table 1, and the reported exonic variants are depicted in Figure 2.

### 2.1. APOBEC3A and APOBEC3B

APOBEC3A and APOBEC3B have been reported to play a role in the restriction and/or evolution of several viruses, including HIV-1 [86,87,88], HBV [54,89], HPV [62,90], parvoviruses [91], and herpesviruses [63,64,66], as well as endogenous elements including Alu [92], LINE-1 [92,93], and HERVK [94]. These two enzymes localize to the nucleus and are expressed at very low levels in CD4 T-cells [95]. Therefore, their potential inhibitory roles in HIV-1 infection are likely limited to myeloid cells [88]. It is worth noting that APOBEC3A and APOBEC3B are currently considered the main sources of TCW > TT/GW (W: T or A) mutations in many tumors [96,97,98]. The main polymorphism that has been reported for APOBEC3A/B is a ~30 kb deletion, which is in high linkage disequilibrium with its surrogate SNP rs12628403 [99]. This deletion has removed all of the APOBEC3B exons and introns, except for its last exon (exon 8), which is identical to the last exon of APOBEC3A (exon 5) [100]. Genomes with this deletion do not express APOBEC3B mRNA, but they express a fused APOBEC3A/B mRNA, which codes for a protein that is identical to APOBEC3A. About 40% of the world populations are estimated to have at least one copy of this deletion, which is highly enriched in Eastern Asian, Oceanic, and Amerindian populations [100]. The APOBEC3B deletion polymorphism has been mostly studied in the context of HIV and HBV infections, and little is known about the role of this deletion in other viral infections. The only exception is a co-infection study by Prasetyo et al., in which an analysis of 597 HIV-1-positive samples from Indonesian populations showed higher rates of co-infection by HBV, HCV, TTV (torque teno virus), and Toxoplasma gondii in individuals with an APOBEC3B deletion genotype [101]. HBV studies are limited, but overall, they seem to suggest little/no difference in the HBV disease risk among people with and without APOBEC3B. For instance, a study of 724 HBV carriers and 469 healthy subjects showed neither an association between the APOBEC3B deletion and HBV infection status, nor with the extent of viral hypermutation [102]. Similarly, a study of 179 chronic HBV patients and 216 healthy Moroccan donors did not show a difference in the frequency of APOBEC3B deletion between the healthy and disease groups. However, individuals having the deletion genotype showed a faster progression to liver diseases [103].

In contrast with HBV, data on HIV-1 are inconsistent. A study of 150 HIV-infected patients and 150 healthy donors in Western India showed a lower risk of HIV-1 acquisition in the donors with no APOBEC3B deletion [104]. Similar results were reported in a study including over 4000 donors from five HIV-1 cohorts. This study, which was conducted using samples from European Americans and African Americans, found no effect associated with the heterozygous APOBEC3B deletion, but reported direct associations between homozygous deletion and HIV-1 acquisition, progression to AIDS, and viral set point [105]. By contrast, other studies have found no evidence for such associations. For example, Itaya et al. analyzed the data from previously reported Japanese and Indian populations [106] and did not observe a difference in the frequencies of the APOBEC3B deletion allele between the HIV-1-infected and control subjects [107]. The analysis of a cohort of Japanese MSM (men who have sex with men), including 248 HIV-1-infected patients and 207 uninfected donors, reached a similar conclusion. The study found no difference between the infected and uninfected men in terms of their APOBEC3B genotype composition. Additionally, the authors reported no difference in the rates of co-infections by HBV, HCV, and syphilis between the two groups. Furthermore, no difference was observed in disease progression parameters, and PBMCs from the infected and uninfected groups were shown to have comparable susceptibility to HIV-1 infection [108]. Overall, studies on the role of the APOBEC3B deletion polymorphism in the susceptibility to HIV acquisition, rates of co-infections, and disease progression have returned different results. Given that APOBEC3B localizes to the nucleus and is not expressed in CD4 T-cells, this enzyme is expected to have little/no impact on the HIV-1 disease status. The reason for the observed associations is not clear.

### 2.2. APOBEC3C

APOBEC3C is expressed at high levels in many tissues and cell types [95,109], and in vitro studies have shown that it can partially inhibit LINE-1 retrotransposition [110,111]. This enzyme localizes to the nucleus, and, except for its rare variant I188, it has weak/no antiviral activity against exogenous viruses. A change from serine (S) to isoleucine (I) in this position (S188I), which is frequent in sub-Saharan African populations, enhances the anti-HIV activity of APOBEC3C in vitro [112,113]. Nevertheless, compared to other APOBEC3 enzymes, such as APOBEC3G and APOBECH, this APOBEC3C variant has limited mutagenic activity. Therefore, it has been postulated that APOBEC3C can potentially play a role in HIV-1 diversification [113].

### 2.3. APOBEC3D

There are not many coding variations within APOBEC3D; therefore, studies focusing on APOBEC3D variations are limited. There is currently no crystal structure available for this enzyme. APOBEC3D localizes to the cytosol and is capable of hypermutating the HIV-1 genome; however, its deaminase activity is limited compared to APOBEC3F, APOBEC3G, and the stable haplotypes of APOBEC3H [9,37,114,115,116]. This enzyme has been suggested to play a role in HIV-1 diversification [37]. Matume et al. analyzed the APOBEC3 polymorphism in 192 HIV-1-infected patients in Northern South Africa, and compared the genotype composition of this cohort with those of the HapMap, 1000 Genomes Project, and ExAC donors [80]. The study identified several variants of APOBEC3D, APOBEC3F, APOBEC3G, and APOBEC3H. The APOBEC3D variants R97C and T238A were reported to be more frequent in the HIV-1-infected cohort than in the African population of the 1000 Genomes Project. Importantly, a study by Duggal et al. showed that the R97C and R248K variants of APOBEC3D have significantly lower antiviral activities against HIV-1Δvif [117]. The study also showed that these two variants, which are specific to sub-Saharan African populations, were highly sensitive to HIV-1 Vif. The variant R248K also showed decreased anti-Alu activity; however, no anti-Alu activity difference was found for R97C. The authors associated this global reduction in the antiviral activity of the R248K mutant to its reduced expression [117].

### 2.4. APOBEC3F

Human APOBEC3F has been shown to inhibit HIV-1 via both deaminase and deaminase-independent mechanisms, although there is growing evidence suggesting the latter is the primary mechanism of viral inhibition [118,119,120,121,122,123]. APOBEC3F is less mutagenic compared to APOBEC3G, and can induce HIV-1 evolution and drug escape [35,37,124]. Not much is known about the role of this enzyme in inhibiting other viruses. A study by Matume et al. showed that the APOBEC3F variants A108S and Y307C were more frequent in a cohort of HIV-1-infected patients in Northern South Africa than the African populations of the 1000 Genomes Project [80]. To investigate the role of APOBEC3F variations in the susceptibility to HIV-1 acquisition and disease progression, An et al. performed a large-scale association analysis, using data from six pretreatment HIV/AIDS cohorts (n = 4203) [125]. The study included 707 European American and 281 African American incident seroconverters, 1135 uninfected but at risk individuals, and 2076 previously infected donors. The presence of valine (V) instead of isoleucine (I) in APOBEC3F position 231 (rs2076101) was reported to be associated with a lower set-point viral load, slower progression to AIDS, and delayed development of pneumocystis pneumonia [125]. In vitro experiments suggested that variations in this amino acid position influence the HIV-1 Vif-mediated degradation of APOBEC3F [125]. In a separate study, the variant 231V was shown to increase HIV-1 restriction by APOBEC3F. The authors suggested that the effect was due to an increase in APOBEC3F steady-state levels, and, therefore, partial protection of APOBEC3F from degradation mediated by HIV-1 Vif [126]. Importantly, this variant is highly abundant in African populations and data suggest it has an increased level of viral encapsidation [126]. Overall, rs2076101 (V231I) seems to be the only APOBEC3F SNP whose differential protective effect is reported in different studies. A study by Lassen et al. has shown that APOBEC3F mRNA splicing also plays a role in the antiviral activity of this enzyme [84]. Using PBMCs from ten healthy donors, the study identified two APOBEC3F splice variants with different expression levels in different cell types. One isoform, which is resistant to Vif, lacks exon 2 (APOBEC3F Δ2) and is expressed in macrophages and monocytes. The other isoform lacks exons 2, 3, and 4 (APOBEC3F Δ2–4) and is very sensitive to Vif-mediated degradation. Compared to the main APOBEC3F isoform, these two splice variants showed reduced viral encapsidation, antiviral activity, and mutagenicity [84].

### 2.5. APOBEC3G

The majority of hypermutated HIV-1 sequences have a GG > AG mutation signature, which is attributed to APOBEC3G. This enzyme is a potent antiviral protein and therefore has been studied by many investigators in the field (reviewed in [1,4,8,127]). APOBEC3G also inhibits other viruses, such as HBV [128,129], HTLV-1 [58,59], and Alu [130]. Several coding and non-coding variants have been reported for this enzyme, however, the most studied variation is H186R (rs8177832). It has been shown that the variant 186R has lower antiviral activity compared to 186H [131]. The variant 186H is prevalent outside of Africa, and 186R is mostly present in sub-Saharan African populations. An et al. genotyped 3073 donors from six HIV–AIDS cohorts and found seven SNPs, including H186R [79]. Three of these SNPs are upstream of APOBEC3G, with potential regulatory roles (-571G/C (rs5757463), -199G/A (rs34550797), and -90C/G (rs5750743)), two are within introns (197193T/C (rs3736685) and 199376G/C (rs2294367)), and two are within exons (F119F (rs5757465) and H186R (rs8177832)). The 186R variant was found to be associated with a decline in the number of CD4 T-cells and an accelerated progression to AIDS [79]. These findings were confirmed in a study of 1049 children with symptomatic HIV-1 infection, who were enrolled in Pediatric AIDS Clinical Trial Group (PACTG) [132]. The participants in that study were 60% black, 26% Hispanic, 13% white, and 1% unknown ethnicity. The APOBEC3G homozygous 186R/R variant was reported to be associated with rapid HIV-1 disease progression and central nervous system impairment. Additionally, the C allele of F119F was found to be associated with protection against disease progression [132]. However, a study focused on a French cohort of 327 HIV-1 patients with extreme disease progression phenotypes and 446 healthy subjects did not show such associations, neither for H186R nor for any of the other reported SNPs [133]. This association was also not observed in a study conducted in Montreal, on samples from Caucasians exposed to HIV-1 [134]. A total of 69 individuals infected with HIV-1, by homosexual contact or drug injection, and 53 exposed seronegative individuals from the same exposed cohort were included in that study. The authors identified several SNPs in their cohort, but none except for an intronic SNP (C40693T, rs17496018) was associated with an increased risk of HIV-1 infection. The variant Q275E (rs17496046) was among the SNPs that did not show an association in Caucasians [134]. By contrast, Matume et al. reported that Q275E is more frequent in a cohort of HIV-1-infected patients in Northern South Africa than in the African populations of the 1000 Genomes Project [80]. The polymorphisms H186R and C40693T (rs17496018) were also studied in two papers by De Maio et al., using samples from 534 children who were perinatally exposed to HIV-1 [135,136]. The studies did not identify an association between the APOBEC3G polymorphism and the transmission or progression to pediatric AIDS. Also, no association was found between APOBEC3G variations and the level of G > A mutations in viral genomes. However, C40693T (rs17496018) was reported to be associated with changes in the Vif motifs that interact with APOBEC3G.

The APOBEC3G coding variation H186R has also been studied by several other groups, but, overall, the results are inconsistent. For example, Singh et al. compared 153 HIV patients with 156 healthy controls in Western India and found no homozygous RR donors in either group [137]. The study found 13% H/R heterozygosity in the healthy donor group and none in the HIV group. This is different from a study involving North Indian donors, which did not find a single 186R variant among a total of 560 subjects studied (50 HIV-1-exposed seronegative donors, 320 HIV-1 seronegative healthy subjects, and 190 HIV-1 seropositive patients) [138]. The observed discrepancies might be due to differences in the studied populations and the overall low frequencies of minor variants. Most of the APOBEC3G variants, including H186R and Q275E, are highly population-specific and their minor variants are mostly found in people with a sub-Saharan African origin. The frequencies of these minor variants are negligible in other populations [132]. This raises the possibility that studies focusing on non-African populations might not have enough statistical power to detect associations between these SNPs and disease status.

In a study by Reddy et al., the role of APOBEC3G expression and polymorphisms in HIV-1 susceptibility and early disease pathogenesis was investigated using samples from 250 South African women who were at high risk of HIV-1 infection [139]. The authors found that within the first 12 months of infection, HIV-positive donors have significantly lower levels of APOBEC3G mRNA than HIV-negative donors. This trend was also observed when within-patient pre- and post-infection samples were analyzed. However, the study found no association between APOBEC3G mRNA levels and viral loads or CD4 T-cell counts in the 32 seroconverters studied, or in the donors who were consistently seronegative. Among the twenty-four APOBEC3G genetic variants identified in this cohort, rs8177832 (H186R) and a downstream variant rs35228531 (C > T) showed strong associations with high viral loads and low CD4 T-cell counts [139]. Compaore et al. also investigated three of these APOBEC3G variants, but using a cohort of 336 seropositive and 372 seronegative donors from Burkina Faso [140]. It was shown that these variants are genetically linked and the haplotype GGT (rs6001417, rs8177832 (H186R), and rs35228531, respectively) has a protective effect. By contrast, the haplotypes GGC and CGC were associated with an increased risk of HIV-1 infection. The same team also studied the role of these variants in HBV/HIV co-infection [141]. The T minor allele of rs35228531 was found to be protective against HBV/HIV co-infection, and among the reported haplotypes, only GGT was protective. By contrast, in a study of 179 HBV chronic carriers and 216 healthy donors in Morocco, the polymorphism H186R was shown to be irrelevant in HBV acquisition [103].

In addition to the above-mentioned coding SNPs and their linked non-coding variants, several other SNPs within APOBEC3G introns and its upstream/downstream regions have been reported. Again, the results appear to be population-dependent. For example, a study of 400 therapy-naïve HIV-1-infected patients from Brazil showed that the CC homozygous genotype of an SNP upstream of APOBEC3G (rs5757463, C/G) was associated with an increased CD4 T-cell count. The donors with CG and GG genotypes showed lower CD4 T-cell counts. The study did not find any other associations in the APOBEC3G locus [142]. This SNP and another upstream SNP (rs5750743, C/G) were investigated in 153 HIV-1 patients and 156 healthy donors from India. It was shown that the heterozygous CG genotypes for both SNPs were more frequent in the HIV-1 group than the control group. The GG genotype for the SNP rs5750743 was less frequent in the HIV-1 group, and the dominant CG + CC genotype for the SNP rs5757463 occurred at a higher level in the HIV-1 patients [143]. These SNPs and several other noncoding SNPs were also studied in a Zimbabwean pediatric population, to investigate the role of APOBEC3G polymorphisms in the susceptibility to HIV infection among children who were born to HIV-infected mothers [144]. The study, which involved 104 children aged 7–9 years, did not find any associations between those APOBEC3G variations (or their combinations as part of haplotypes) and HIV susceptibility. Finally, an analysis of the association between the APOBEC3 polymorphism and HIV-1 subtype B hypermutation in a West Australian HIV cohort (n = 127) showed a marginal association for the intronic SNP rs5757467 [145].

### 2.6. APOBEC3H

APOBEC3H is the most variable member of the APOBEC3 family. This protein has the following five major non-synonymous variations: N15Δ (rs140936762), R18L (rs139293), G105R (rs139297), K121D (rs139299 and rs139298), and E178D (rs139302), collectively coding for at least 13 protein haplotypes with diverse geographic distributions (Figure 3A,B). Four of these haplotypes (I–IV) have an allelic frequency of >1%. The haplotypes II (NRRDD), V (NRRDE), and VII (NRRKE) show strong antiviral activities [32,83,85,146,147]. Variants with a deletion in position 15 and/or G in position 105 are unstable and have little/no antiviral activity, likely due to ubiquitination [148]. Differential interactions with HIV-1 Gag have also been suggested to be responsible for the significantly reduced antiviral activity of HapI, which has a G in position 105 [149]. APOBEC3H HapII interacts with the nucleocapsid in an RNA-dependent manner, but APOBEC3H HapI interacts with the C-terminus of the matrix and the N-terminus of capsid proteins. Therefore, incorporation into the viral core is essential for the antiviral activity of APOBEC3H [149]. Nuclear localization of APOBEC3H HapI has also been suggested to explain its reduced antiviral activity [81].

Gourraud et al. genotyped APOBEC3H in 96 HIV-1-infected therapy-naïve patients, and showed that a cluster of SNPs (rs139316, rs139317, rs139323, and rs139336), which are genetically linked to the known SNPs G105R, K121D, and E178D are associated with differences in GA > AA hypermutation and viral RNA levels [150]. It is known that the HIV-1 Vif amino acid positions 39 and 48 play a key role in APOBEC3H counteraction, and changes in these two positions are linked to the APOBEC3H genotype of patients [26,27,28,151,152]. Ooms et al. determined the APOBEC3H haplotypes and HIV-1 Vif sequences in 76 infected patients, and found higher viral loads and lower CD4 T-cell counts in patients who were homozygous for unstable APOBEC3H haplotypes. Their study showed an enrichment of phenylalanine (F) at the Vif amino acid position 39 in donors with a stable APOBEC3H haplotype, compared to those with unstable APOBEC3H variants. A mutation of phenylalanine (F) to valine (V) abolished the ability of Vif to counteract APOBEC3H [27]. The Vif variants 39F and 48H can degrade stable APOBEC3H enzymes, and are referred to as ‘hyper-Vif’. Other variants (39 V and 48 N), which do not counteract APOBEC3H, are known as ‘hypo-Vif’ [26,27]. Refsland et al. used hyper-Vif and hypo-Vif variants of HIV-1 to infect primary cells with different APOBEC3H haplotypes. All the viruses replicated in cells with unstable APOBEC3H haplotypes I, III, and IV, but only hyper-Vif HIV-1 showed robust replication in cells with stable APOBEC3H haplotypes (Haps II and V). Importantly, the study found a positive correlation between the prevalence of hyper-Vif HIV-1 isolates and the allelic frequency of APOBEC3H stable haplotypes across the globe [26]. This selective pressure on the HIV-1 Vif, imposed by APOBEC3H, has been confirmed in a humanized mouse model [28]. Altogether, these data suggest that HIV Vif adapts to stable and restrictive variants of APOBEC3H. Interestingly, a study by Benito et al. reported that the frequencies of unstable APOBEC3H haplotypes (those with 15Δ and 105G) were lower in HIV-1 elite controller (EC) patients than non-controller (NC) donors (23/30 = 77% EC vs. 10/11 = 91% NC) [153]. The overall high percentages of unstable APOBEC3H haplotypes in this Spanish cohort is not unexpected, because the stable haplotypes of APOBEC3H (II, V, and VII) are mostly found in people of sub-Saharan African origin. Two studies focusing on Japanese and Indian populations showed that the variations 15Δ and 105G, which make APOBEC3H unstable, are associated with the susceptibility to HIV-1 infection [154,155]. Additionally, a study by Matume et al. showed that the APOBEC3H variants N15Δ, R18L, G105R, and E178D are found more frequently in a cohort of HIV-1-infected patients in Northern South Africa compared to the African populations of the 1000 Genomes Project [80].

Four splice variants (SV200/183/182/154) have been reported for APOBEC3H, three of which (SV200, SV183, and SV182) show antiviral activity [32,83] (Figure 3A). Harari et al. analyzed the PBMCs from 12 healthy donors, and showed that all of the APOBEC3H haplotypes I–IV were resistant to Vif. The authors tested the antiviral and deaminase activities of different APOBEC3H splice variants using haplotypes I and II expression vectors, and found that all of the HapII splice variants were significantly more antiviral than those of HapI. Overall, HapII SV200 showed slightly more antiviral and deaminase activities than Hap II SV182/3 [83]. Gu et al. expanded this work, and they studied all four splice variants and seven haplotypes I-VII of APOBEC3H. The haplotypes II, V, and VII were shown to have high cytosine deamination activity. Additionally, it was shown that these haplotypes, particularly HapII, also target methylated cytosines for deamination. The deaminase activity of these variants were correlated with their anti-HIV activity [85]. Gu et al. only investigated the HapI version of the APOBEC3H splice variants, and observed that HapI SV200 had lower enzymatic activity compared to other HapI isoforms. Interestingly, on a HapII backbone, this trend is reversed, and HapII SV200 shows the highest antiviral activity [32,83]. Data on the role of these splice variants in inhibiting other viruses are limited, but HapII SV183 has been shown to have the highest antiviral activity against HBV [55].

In addition to exonic variations, *APOBEC3H* has an intronic indel polymorphism (CTC/Δ, rs149229175) that alters mRNA splicing [32]. Similar to other *APOBEC3H* genetic variations, this SNP is highly population-specific. The CTC deletion is almost exclusively embedded in the HapII genome, which is present (at least one copy) in ~80% of sub-Saharan African populations. The CTC deletion induces the inclusion of an intronic antisense L1 element as a new exon (exon 4b) in APOBEC3H HapII mRNA. The result of this L1 exonization is a new splice variant (SV200) with an extended and different C-terminus. Importantly, the C-terminus tail of SV200 is cleaved by the HIV-1 protease, to form a new variant with 186 amino acids (SV186), with reduced antiviral activity [32] (Figure 3C).

## 3. Future Directions and Conclusions

The studies of APOBEC3 variations have mostly focused on genetic variations. Numerous polymorphic sites have been reported for APOBEC3 enzymes, particularly for APOBEC3G, which is known for its potent anti-HIV activity, and APOBEC3H, which is known for its high diversity. Variations in both the APOBEC3 gene bodies and regulatory regions have been shown to play a role in the biology of APOBEC3 enzymes and their interaction with viruses such as HIV-1 and HBV. However, the molecular mechanisms underlying the observed differences associated with APOBEC3 genetic variations are mostly unknown. For example, it is not clear why the carriers of the APOBEC3G 186R variant progress to AIDS faster. The evolutionary origin of APOBEC3 variations and the selection forces diversifying APOBEC3 enzymes in different human populations are also mostly unknown. As an example, it is unclear what led to APOBEC3H diversification, to the extent that three of its main haplotypes, Haps I, III, and IV, lost their antiviral activities. These unstable haplotypes account for ~77% of all APOBEC3 variants in human populations, yet little is known about their roles. Strikingly, the APOBEC3H haplotypes III and IV, despite being genetically similar, have widely different geographic distributions. APOBEC3H HapIII is prevalent in sub-Saharan African populations and APOBEC3H HapIV is mostly found outside of Africa. This might be to a genetic bottleneck, or possibly a selection mechanism, which is yet to be studied and identified. APOBEC3 genes are coded in tandem on chromosome 22, and form haplotype blocks extending across two or more *APOBEC3* genes. Also, due to their close proximity, a considerable amount of splicing occurs between the exons of adjacent APOBEC3 genes. Currently, little is known about these APOBEC3 genomic and transcriptomic interdependences, and their roles in viral hypermutation and inhibition. Mutation of viruses by APOBEC3 enzymes can have widely different functional consequences, depending on the genetic makeup of both the patient and the virus. Too much cytosine deamination by potent APOBEC3 enzymes, such as APOBEC3G and APOBEC3H HapII, can be lethal to the virus. This is particularly true for stop codons, which are generated at a higher rate by APOBEC3G (GG > AG) compared to other APOBEC3 enzymes (GA > AA). By contrast, sublethal mutations have been shown to drive viral diversification and drug escape. It is not clear what APOBEC3 enzymes and variants are responsible for lethal versus sublethal viral mutations, and whether these play a role in donor-specific and/or and population-specific responses to infection and antiviral therapies. Most of these key questions are cross-disciplinary by nature, and they require extensive collaborations among wet-lab and quantitative scientists to answer. More importantly, they require data sharing, something that is still not practiced by many scientists in the field. It is very unfortunate that, despite many datasets generated from studies of elite controller HIV-1 patients, little is publicly available. Overall, there is a lot to learn about differential antiviral immunity in humans by studying APOBEC3 variations, and this provides ample opportunities for research and training across quantitative and experimental biosciences.

## Figures and Tables

**Figure 1 viruses-13-01366-f001:**
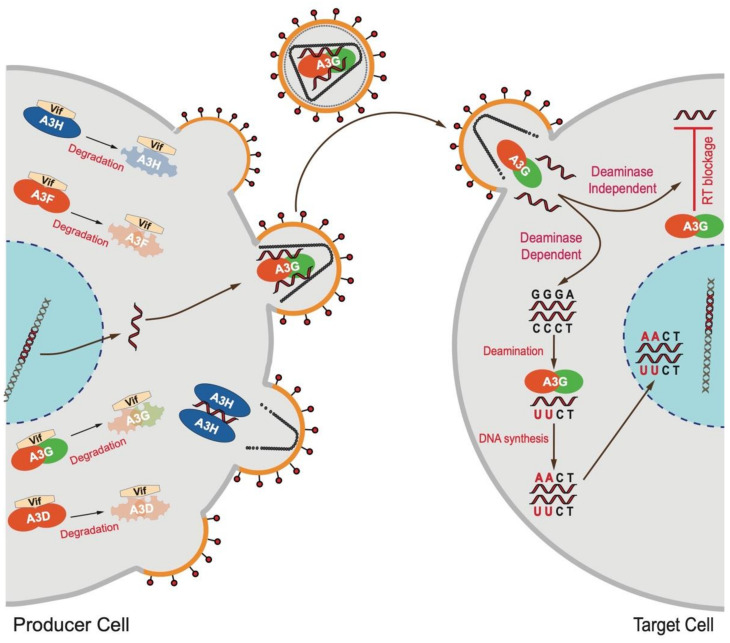
Inhibition of HIV-1 by APOBEC3 enzymes using deaminase-dependent and -independent mechanisms. APOBEC3 enzymes are often counteracted by the HIV-1 Vif protein. Those not counteracted by Vif can package into the budding virions and inhibit viral replication by inflicting C-to-U hypermutation in the HIV-1 cDNA or by deaminase-independent mechanisms such as the physical blockage of HIV-1 reverse transcription.

**Figure 2 viruses-13-01366-f002:**
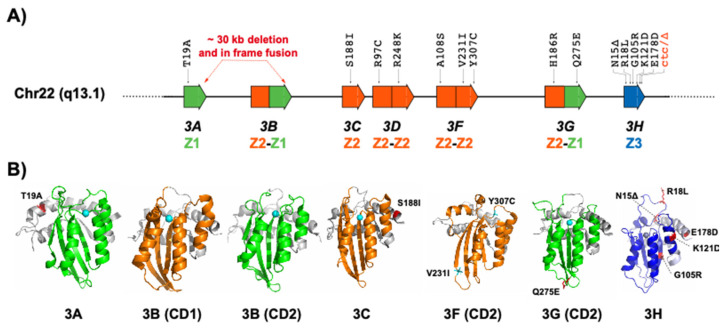
A schematic of main human APOBEC3 variantions shown on (**A**) human chr22 and (**B**) the available human APOBEC3 structures. The catalytic domains Z1, Z2, and Z3 are colored in green, orange, and blue, respectively.

**Figure 3 viruses-13-01366-f003:**
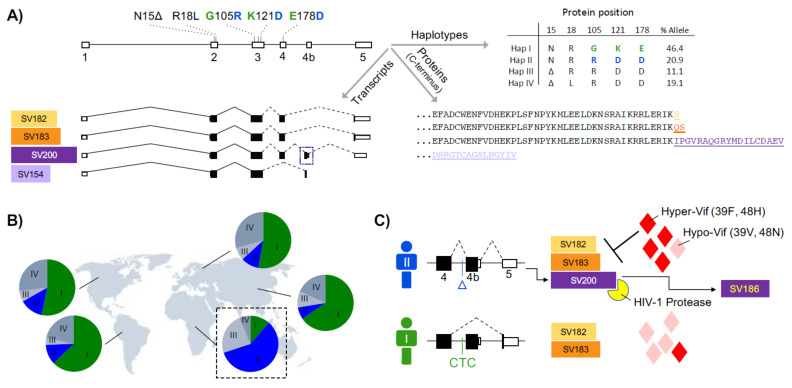
APOBEC3H variations. (**A**) Main APOBEC3H haplotypes, splice variants, and protein variants; (**B**) geographic distribution of main APOBEC3H haplotypes; (**C**) a schematic of population-specific APOBEC3H mRNA splicing, HIV-1 protease processing of SV200 into SV186, and adaptation of HIV-1 Vif to HapII.

**Table 1 viruses-13-01366-t001:** Summary of APOBEC3 variations and their effects.

	Description	Allelic Frequency			Studied Populations and References	Effect
		African	European	Asian		
APOBEC3A/B						
rs12628403 (surrogate for ~30 kb APOBEC3B deletion)	~30 kb deletion (Δ) of all APOBEC3B exons and introns except for exon 8; In-frame fusion to APOBEC3A.	n = 3394 Δ = 0.03	n = 20,404 Δ = 0.08	n = 116 Δ = 0.4	Indonesian PMID: 26305823	HIV-1 co-infections: Δ > reference
					Moroccan PMID: 24010642	%Δ: HBV patients ~ healthy donors Progression to liver diseases: Δ > reference
					Western Indian PMID: 27522954	Risk of HIV-1 acquisition: Δ > reference
					European Americans and African Americans PMID: 19698078	HIV-1 acquisition, progression to AIDS, and viral set point: Δ/Δ > reference
					Japanese and Indian PMID: 20684727	%Δ: HIV-1 patients ~ healthy donors
					Japanese PMID: 19788695	%Δ: HBV patients ~ healthy donors Hypermutation context: Δ ~ reference
					Japanese PMID: 24667791	%Δ: HIV-1 patients ~ healthy donors HIV-1 co-infections: Δ ~ reference HIV-1 progression: Δ ~ reference
APOBEC3C						
rs112120857	Missense (G > C / G > T) S188I	n = 3574 G = 0.94 C = 0.00 T = 0.06	n = 37,290 G = 0.99962 C = 0.00003 T = 0.00035	n = 168 G = 1.00 C = 0.00 T = 0.00	PMIDs: 27732658, 30558640	In vitro anti-HIV-1 activity: I > S
APOBEC3D						
rs201709403	Missense (A > G) T238A	n = 2898 A = 1.00 G = 0.00	n = 9690 A = 1.00 G = 0.00	n = 112 A = 1.00 G = 0.00	Northern South African PMID: 30660178	%G (238A): HIV-1 patients > 1000 Genomes Africans
rs75858538	Missense (C > T) R97C	n = 5146 C = 0.98 T = 0.02	n = 180,258 C = 0.9999 T = 0.0001	n = 6364 C = 0.9998 T = 0.0002	Northern South African PMID: 30660178	%T (97C): HIV-1 patients > 1000 Genomes Africans
					PMID: 23755966	Protein expression: R ~ C Anti-HIV-1 activity: R > C Sensitivity to HIV-1 Vif: R ~ C Anti-Alu activity: R ~ C
rs61748819	Missense (G > A) R248K	n = 3574 G = 0.91 A = 0.09	n = 32,834 G = 0.9997 A = 0.00034	n = 168 G = 1.00 A = 0.000	PMID: 23755966	Protein expression: R > K Anti-HIV-1 activity: R > K Sensitivity to HIV-1 Vif: R ~ K Anti-Alu activity: R > K
APOBEC3F						
rs2020390	Missense (G > T) A108S	n = 3512 G = 0.60 T = 0.40	n = 37,024 G = 0.49 T = 0.51	n = 168 G = 0.29 T = 0.71	Northern South African PMID: 30660178	%T (108S): HIV-1 patients > 1000 Genomes African populations
rs12157816	Missense (A > G) Y307C	n = 3574 A = 0.97 G = 0.03	n = 37,264 A = 0.9881 G = 0.0119	n = 168 A = 1.0 G = 0.0	Northern South African PMID: 30660178	%G (307C): HIV-1 patients > 1000 Genomes African populations
rs2076101	Missense (G > A) V231I	n = 8220 G = 0.73 A = 0.27	n = 256,418 G = 0.48 A = 0.52	n = 3812 G = 0.3 A = 0.7	European American and African American PMID: 26942578	Set-point viral load: I > V Progression to AIDS: I > V Pneumocystis pneumonia: I > V
					PMID: 30448640	HIV-1 restriction: V > I Protection against HIV-1 Vif: V > I Level of viral encapsidation: V > I
APOBEC3FΔ2	Isoform lacking exon 2				PMID: 20624919	Sensitivity to Vif: main isoform > APOBEC3FΔ2 Viral packaging: main isoform > APOBEC3FΔ2 Antiviral activity: main isoform > APOBEC3FΔ2 Deamination: main isoform > APOBEC3FΔ2
APOBEC3FΔ2-4	Isoform lacking exons 2, 3 and 4				PMID: 20624919	Sensitivity to Vif: APOBEC3FΔ2–4 > main isoform Viral packaging: main isoform > APOBEC3FΔ2–4 Antiviral activity: Main isoform > APOBEC3FΔ2–4 deamination: main isoform > APOBEC3FΔ2–4
APOBEC3G						
rs5757463	-571G/C upstream (G > C)	n = 106 G = 0.29 C = 0.71	n = 6564 G = 0.06 C = 0.94	n = 4 G = 0.0 C = 1.0	African Americans and European Americans PMID: 15452227	Susceptibility to HIV-1 infection: G ~ C
					Brazilian PMID: 20874421	CD4 T-cell count: CC > CG and GG
					West Indian PMID: 29864532	%CG: HIV patients > healthy donors %(GC + CC): HIV patients > healthy donors
					Zimbabwean PMID: 27245545	Susceptibility to HIV-1 infection: G ~ C
rs34550797	-199G/A upstream (G > A)	n = 2816 G = 0.995 A = 0.005	n = 7618 G = 0.9992 A = 0.0008	n = 108 G = 1.0 A = 0.0	African Americans and European Americans PMID: 15452227	Susceptibility to HIV-1 infection: G ~ A
rs5750743	-90C/G upstream/5′UTR (C > G)	n = 276 C = 1.00 G = 0.00	n = 5330 C = 0.54 G = 0.46	n = 8 C = 1.0 G = 0.0	African Americans and European Americans PMID: 15452227	Susceptibility to HIV-1 infection: C ~ G
					West Indian PMID: 29864532	%CG: HIV-1 patients > healthy donors %GG: healthy donors > HIV patients
					Zimbabwean PMID: 27245545	Susceptibility to HIV-1 infection: C ~ G
rs5757465	Synonymous (T > C) F119F	n = 11,214 T = 0.90 C = 0.10	n = 248,594 T = 0.56 C = 0.44	n = 3908 T = 0.76 C = 0.24	African Americans and European Americans PMID: 15452227	Susceptibility to HIV-1 infection: T ~ C
					Diverse populations PMID: 23138837	HIV-1 disease progression: T > C CNS impairment: T > C
rs3736685	197193T/C intron (T > C)	n = 7982 T = 0.63 C = 0.37	n = 223,498 T = 0.97 C = 0.03	n = 3870 T = 0.92 C = 0.08	African Americans and European Americans PMID: 15452227	Susceptibility to HIV-1 infection: T ~ C
					Zimbabwean PMID: 27245545	Susceptibility to HIV-1 infection: T ~ C
rs2294367	199376G/C intron (C > G)	n = 2036 C = 0.99 G = 0.01	n = 8254 C = 0.57 G = 0.43	n = 16 C = 0.75 G = 0.25	African Americans and European Americans PMID: 15452227	Susceptibility to HIV-1 infection: C ~ G Progression to AIDS: G > C
					Zimbabwean PMID: 27245545	Susceptibility to HIV-1 infection: C ~ G
rs8177832	Missense (A > G) H186R	n = 11,678 A = 0.66 G = 0.34	n = 252,598 A = 0.97 G = 0.03	n = 6820 A = 0.92 G = 0.08	African Americans and European Americans PMID: 15452227	Susceptibility to HIV-1 infection: H ~ R Rate of CD4 T cell loss: R > H Progression to AIDS: R > H
					PMID: 27064995	Antiviral activity: H > R Counteraction by Vif: H ~ R
					Diverse populations PMID: 23138837	HIV-1 disease progression: R > H CNS impairment: R > H
					South Africans PMID: 19996938	Viral load: R > H CD4 T cell count: H > R
					French PMID: 15609224	Progression to AIDS: H ~ R
					Caucasians PMID: 16988524	Progression to AIDS: H ~ R
					Argentinian PMID: 22145963	Progression to AIDS: H ~ R Viral G-to-A mutation: H ~ R
					Argentinian PMID: 21571098	HIV-1 transmission: H ~ R Progression to AIDS: H ~ R Viral G-to-A mutation: H ~ R
					Zimbabwean PMID: 27245545	Susceptibility to HIV-1 infection: H ~ R
					Morocco PMID: 24010642	Risk of HBV acquisition: H ~ R
					Burkina Fasoian PMID: 26741797	Protection against HIV-1 infection: GGT [rs6001417, rs8177832 (H186R), rs35228531] > Other haplotypes Risk of HIV-1 infection: GGC and CGC > Other haplotypes
					Burkina Fasoian PMID: 27449138	HIV-1/HBV co-infection rate: other haplotypes > GGT [rs6001417, rs8177832 (H186R), rs35228531]
					Western Indian PMID: 26853443	Risk of HIV-1 acquisition: H ~ R Progression to AIDS: H ~ R
					North Indian PMID: 18652534	Absence of 186R variant
rs17496018	C40693T Intron (C > T)	n = 4296 C = 0.94 T = 0.06	n = 26,952 C = 0.93 T = 0.07	n = 128 C = 0.97 T = 0.03	Caucasians PMID: 16988524	Risk of HIV-1 infection: T > C
					Argentinian PMID 22145963	Progression to AIDS: H ~ R Viral G > A mutation: H ~ R
					Argentinian PMID: 21571098	HIV-1 transmission: H ~ R Progression to AIDS: H ~ R Viral G-to-A mutation: H ~ R C > T polymorphism was associated with Vif variation.
rs17496046	Missense (C > G) Q275E	n = 4374 C = 0.87 G = 0.13	n = 96,948 C = 0.93 G = 0.07	n = 3326 C = 0.97 G = 0.03	Caucasians PMID: 16988524	Risk of HIV-1 infection: C ~ G
					Northern South African PMID: 30660178	% G (275E): HIV-1 patients > 1000 Genomes Africans
rs6001417	Intron (C > G)	n = 2960 C = 0.63 G = 0.37	n = 16,474 C = 0.97 G = 0.03	n = 112 C = 0.94 G = 0.06	Burkina Fasoian PMID: 26741797	Protection against HIV-1 infection: GGT [rs6001417, rs8177832 (H186R), rs35228531] > Other haplotypes Risk of HIV-1 infection: GGC and CGC > Other haplotypes
					Burkina Fasoian PMID: 27449138	HIV-1/HBV co-infection rate: other haplotypes > GGT [rs6001417, rs8177832 (H186R), rs35228531]
rs35228531	Downstream (C > T)	n = 1564 C = 0.994 T = 0.006	n = 9814 C = 1.00 T = 0.00	n = 112 C = 1.00 T = 0.00	South African PMID: 19996938	Viral load: T > C CD4 T-cell count: C > T
					Burkina Fasoian PMID: 26741797	Protection against HIV-1 infection: GGT [rs6001417, rs8177832 (H186R), rs35228531] > Other haplotypes Risk of HIV-1 infection: GGC and CGC > Other haplotypes
					Burkina Fasoian PMID: 27449138	HIV-1/HBV co-infection rate: C > T HIV-1/HBV co-infection rate: Other haplotypes > GGT [rs6001417, rs8177832 (H186R), rs35228531]
rs5757467	Intron (C > T)	n = 2946 C = 0.31 T = 0.69	n = 15,414 C = 0.36 T = 0.64	n = 112 C = 0.21 T = 0.79	West Australian PMID: 16940537	HIV-1 hypermutation: C > T
APOBEC3H						
rs139292 (Previously rs140936762)	Indel (CAA/Δ) N15Δ	n = 3506 Δ = 0.32	n = 20,264 Δ = 0.34	n = 168 Δ = 0.23	PMID: 32235597	Antiviral activity: Reference > Δ
					Northern South African PMID: 30660178	%Δ: HIV-1 patients >1000 Genomes Africans
					Japanese PMID: 25721876	Susceptibility to HIV-1 infection: Δ > reference
					Indian PMID: 26559750	Susceptibility to HIV-1 infection: Δ > reference
rs139293	Missense (G > T) R18L	n = 3158 G = 0.96 T = 0.04	n = 36,146 G = 0.75 T = 0.25	n = 166 G = 0.92 T = 0.08	Northern South Africa PMID: 30660178	%T(18L): HIV-1 patients > 1000 Genomes Africans
rs139297	Missense (G > C) G105R	n = 1308 G = 0.66 C = 0.34	n = 35,022 G = 0.56 C = 0.44	n = 148 G = 0.89 C = 0.11	PMID: 32235597	Ubiquitination: G (HapI) > R (HapII)
					PMID: 20519396	Antiviral activity: R (HapII) > G (HapI) HIV-1 encapsidation: R (HapII) ~ G (HapI) Protein expression: R (HapII) > G (HapI) Interaction with Gag: R (HapII) with nucleocapsid; G (HapI) with C-terminal of matrix and N-terminal of capsid
					Japanese PMID: 25721876	Susceptibility to HIV-1 infection: G (HapI) > R (HapII) Progression to AIDS: G (HapI) > R (HapII)
					Indian PMID: 26559750	%15Δ-105R: HIV-1 patients > healthy donors
					Diverse populations PMID: 21167246	Rate of HIV-1 GA-to-AA mutation: R (HapII) > G (HapI)
rs139299	Missense (G > C) K121D	n = 3574 G = 0.24 C = 0.76	n = 37,158 G = 0.52 C = 0.48	n = 168 G = 0.72 C = 0.28	Diverse populations PMID: 21167246	Rate of HIV-1 GA-to-AA mutation: D (HapII) > K (HapI)
rs139298	Missense (A > G) K121D	n = 11,188 A = 0.21 G = 0.79	n = 252,052 A = 0.53 G = 0.47	n = 6744 A = 0.69 G = 0.31	Diverse populations PMID: 21167246	Rate of HIV-1 GA-to-AA mutation: D (HapII) > K (HapI)
rs139302	Missense (G > C) E178D	n = 3882 G = 0.26 C = 0.74	n = 97,472 G = 0.54 C = 0.46	n = 3326 G = 0.67 C = 0.33	PMID: 21167246	Rate of HIV-1 GA-to-AA mutation: D (HapII) > E (HapI)
					Northern South African PMID: 30660178	%C (178D): HIV-1 patients > 1000 Genomes Africans
					PMID: 27534815	Cytosine deamination: E (HapV) > D (other haplotypes) Methyl cytosine deamination: D (HapII) >> E (other haplotypes) DNA binding: all haplotypes are the same.
rs149229175	Indel (CTC/Δ) intron	n = 3410 Δ = 0.32	n = 18,142 Δ = 0.13	n = 164 Δ = 0.08	Diverse populations PMID: 30297863	%SV200: Δ > CTC
SV200, SV183, SV182, and SV154	Splice variants with different C-terminals				Diverse populations PMID: 18945781	Viral restriction: SV200 (HapII) > other variants
					Diverse populations PMID: 30297863	Antiviral activity: HapII-SV200 > SV182/183 Viral encapsidation: HapII-SV182/183 > SV200 Protease processing: only HapII SV200 L1 fragment in transcript: only HapII SV200
					PMID: 27534815	Cytosine and methyl cytosine deamination: HapI-SV182/183 > SV200 >> SV154
					PMID: 31400856	HBV restriction: HapII SV183 > other variants

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
