# Peer review of "Human APOBEC3 Variations and Viral Infection"

_viruses, 2021, doi:10.3390/v13071366_

Round 1
Reviewer 1 Report
This is a manuscript that covers in considerable detail the known variants of the human APOBEC3 family members and their relationship, if any, with resistance or susceptibility to human viruses, mainly HIV and HBV. While the manuscript is comprehensive and covers many studies conducted across the world, it suffers seriously from a lack of perspective. There are many aspects of virus-APOBEC3 interactions that must be explained before the apparent virus susceptibilities/sensitivities can be understood. In the absence of any background information about the structures of the different APOBEC3s, a description of how they interact with viruses and what the changes in the APOBEC3 proteins do to their activity, it is difficult to understand why certain genetic variants occur more frequently in certain infections.
For example, it is not explained anywhere that while some APOBEC3s consist of two similar, but non-identical parts- N-terminal and C-terminal domains-, others have a single domain. Furthermore, the two domains in proteins such as A3B and A3G have very different antiviral functions. Without laying down such a foundation, the reader may not understand that "APOBEC3F variants A108S and Y307C" have changes in respectively, NTD and CTD.
The same lack of explanation exists when it comes to the viruses. While a HIV is an RNA virus, HBV is a DNA virus. This changes the mode of action of APOBEC3s on the viral genomes. Additionally, HIV-1 codes for an antiviral protein, Vif, that makes both A3F and A3G nearly ineffective, no such protein has been described to be coded by HBV. Therefore, one would not expect the same set of mutations in APOBEC3s to be selected in response to infections of HIV-1 and HBV. Furthermore, much is known about Vif-A3G/A3F interaction and the variants of these proteins should be described in the context of this interaction.
Finally, there needs to be a large table that lists all known variants of APOBEC3s for which a biological role such as virus susceptibility has been assigned. Going through discussion of variant after variant is pointless, unless one can look at the whole picture. If a Table that lists the variants, their geographical distribution, their antiviral effects and provide appropriate references, this would be a much more useful review article.
Author Response
REVIEWER 1:
Comments and Suggestions for Authors
This is a manuscript that covers in considerable detail the known variants of the human APOBEC3 family members and their relationship, if any, with resistance or susceptibility to human viruses, mainly HIV and HBV. While the manuscript is comprehensive and covers many studies conducted across the world, it suffers seriously from a lack of perspective. There are many aspects of virus-APOBEC3 interactions that must be explained before the apparent virus susceptibilities/sensitivities can be understood. In the absence of any background information about the structures of the different APOBEC3s, a description of how they interact with viruses and what the changes in the APOBEC3 proteins do to their activity, it is difficult to understand why certain genetic variants occur more frequently in certain infections.
For example, it is not explained anywhere that while some APOBEC3s consist of two similar, but non-identical parts- N-terminal and C-terminal domains-, others have a single domain. Furthermore, the two domains in proteins such as A3B and A3G have very different antiviral functions. Without laying down such a foundation, the reader may not understand that "APOBEC3F variants A108S and Y307C" have changes in respectively, NTD and CTD.
Response: To address this comment, we re-wrote the introduction and included the requested information (lines 24-162). We have also added a new panel to Fig. 2 in which we show all available human A3 structures and variant positions.
The same lack of explanation exists when it comes to the viruses. While a HIV is an RNA virus, HBV is a DNA virus. This changes the mode of action of APOBEC3s on the viral genomes. Additionally, HIV-1 codes for an antiviral protein, Vif, that makes both A3F and A3G nearly ineffective, no such protein has been described to be coded by HBV. Therefore, one would not expect the same set of mutations in APOBEC3s to be selected in response to infections of HIV-1 and HBV. Furthermore, much is known about Vif-A3G/A3F interaction and the variants of these proteins should be described in the context of this interaction.
Response: In the revised manuscript, we have added, in several sections, the information about viruses inhibited by different A3 enzymes and have also highlighted their differences. Please see lines 149-162 as an example.
The focus of this review is APOBEC3 variations only, and we have discussed vif variations such as V39F and N48H, which adapt to APOBEC3H variations. There are many variations in the genomes of HIV-1 and many other viruses with reported links to APOBEC enzymes but not to APOBEC variations. These variations are not within the scope of this review and can be covered in a future review focusing on viral variations.
Finally, there needs to be a large table that lists all known variants of APOBEC3s for which a biological role such as virus susceptibility has been assigned. Going through discussion of variant after variant is pointless, unless one can look at the whole picture. If a Table that lists the variants, their geographical distribution, their antiviral effects and provide appropriate references, this would be a much more useful review article.
Response: Please see Table 1.
Reviewer 2 Report
Summary: In this review article, Sadeghpour et al. seek to provide a comprehensive overview of our current understanding of APOBEC3 genetic variation and its influence on viral replication and pathogenesis. This has been studied by many groups in the past and there have been many conflicting reports, so this review will be a valuable resource for the field moving forward. Overall, the article does a thorough job, referencing a vast majority of genetic studies performed on the APOBEC3 family. That being said, not much effort is done to synthesize the knowledge and sections can read as a laundry list of executive summaries from various publications. Some additional background could also be useful in helping novice readers better digest the material. A few specific recommendations are as follows:
Major Comments:
- The Introduction could use some additional background on exactly which viruses are known to be inhibited by (and counteract) the APOBEC3 family of proteins. The concept of ‘deamination’ should be more formally defined here as the authors use ‘deaminase independent’ terminology throughout the text. Finally, the concept of lethal versus sublethal mutagenesis should be raised here rather than in the Discussion as the authors frequently refer to these enzymes contributing to viral diversification without much context.
- The discussion of each APOBEC3 protein should begin with a very brief overview about what antiviral and other functions have been ascribed to the protein.
- Might we suggest the authors craft a Table with APOBEC3 polymorphisms in the first column, a brief description and Ref for studies that found Significant correlations in the second column, and a brief description and Ref for studies that found NO Significant correlations in the third column. This would bring a lot of clarity to the list of studies provided in the text.
- There are several vague sentences throughout the text that should be refined and made more precise. For example, lines 56-57, states that the “A3B deletion polymorphism has been implicated” in studies, but the number of studies, what kinds of studies, etc. are left undescribed. Several examples follow, but it is up to the reader to summarize and synthesize the information.
- Similarly, there are a few places where jargon and concepts in the APOBEC3 field are mentioned, but not fully discussed. For example, the concept of A3C dimerization in lines 93-95 or the concept of sublethal mutagenesis and diversification in lines 100-101 should be better described so the average readers has a better understanding of their significance.
- The APOBEC3G and APOBEC3H sections are long and unwieldy. We would suggest that these are broken up into multiple paragraphs, each one focusing on a specific polymorphism or haplotype.
- A few grammar and spelling errors persist throughout, though none alter understanding of the text.
- The Conclusion is excellent and the reviewer appreciates the call for additional collaboration in this arena.
Author Response
REVIEWER 2:
Comments and Suggestions for Authors
Summary: In this review article, Sadeghpour et al. seek to provide a comprehensive overview of our current understanding of APOBEC3 genetic variation and its influence on viral replication and pathogenesis. This has been studied by many groups in the past and there have been many conflicting reports, so this review will be a valuable resource for the field moving forward. Overall, the article does a thorough job, referencing a vast majority of genetic studies performed on the APOBEC3 family. That being said, not much effort is done to synthesize the knowledge and sections can read as a laundry list of executive summaries from various publications. Some additional background could also be useful in helping novice readers better digest the material. A few specific recommendations are as follows:
Response: In response to this comment, we have revised the text in several places. See lines 295-298, 353-356, 366-367, 388-389, 903-906, 1002-1003, 1327-1331.
Major Comments:
The Introduction could use some additional background on exactly which viruses are known to be inhibited by (and counteract) the APOBEC3 family of proteins. The concept of ‘deamination’ should be more formally defined here as the authors use ‘deaminase independent’ terminology throughout the text. Finally, the concept of lethal versus sublethal mutagenesis should be raised here rather than in the Discussion as the authors frequently refer to these enzymes contributing to viral diversification without much context.
Response: We have re-written the introduction and have added all the requested information including deaminase versus deaminase-independent mechanisms and lethal versus sub-lethal mutations, and many more (Lines 25-162).
The discussion of each APOBEC3 protein should begin with a very brief overview about what antiviral and other functions have been ascribed to the protein.
Response: Done.
Might we suggest the authors craft a Table with APOBEC3 polymorphisms in the first column, a brief description and Ref for studies that found Significant correlations in the second column, and a brief description and Ref for studies that found NO Significant correlations in the third column. This would bring a lot of clarity to the list of studies provided in the text.
Response: Please see Table 1.
There are several vague sentences throughout the text that should be refined and made more precise. For example, lines 56-57, states that the “A3B deletion polymorphism has been implicated” in studies, but the number of studies, what kinds of studies, etc. are left undescribed. Several examples follow, but it is up to the reader to summarize and synthesize the information.
Response: We changed it to “APOBEC3B deletion polymorphism has been mostly studied in the context of HIV and HBV infections and little is known about its role in other viral infections …”
Similarly, there are a few places where jargon and concepts in the APOBEC3 field are mentioned, but not fully discussed. For example, the concept of A3C dimerization in lines 93-95 or the concept of sublethal mutagenesis and diversification in lines 100-101 should be better described so the average readers has a better understanding of their significance.
Response: The section about dimerization has been removed (Lines 360-362) and lethal vs. sub-lethal mutations scenarios are now covered in detail in the new introduction.
The APOBEC3G and APOBEC3H sections are long and unwieldy. We would suggest that these are broken up into multiple paragraphs, each one focusing on a specific polymorphism or haplotype.
Response: We attempted to use this approach, but ended up with many redundant sections. This is because in most studies more than one variant is analyzed and discussed. However, in response to this comment, we rearranged some of the sections and split the text into paragraphs.
A few grammar and spelling errors persist throughout, though none alter understanding of the text.
Response: We identified multiple typos and fixed them.
The Conclusion is excellent and the reviewer appreciates the call for additional collaboration in this arena.
Response: Thank you.
Reviewer 3 Report
In this review article, the authors, Sadeghpour et al., summarize the association between human APOBEC3 (A3) variations and phenotypes in viral infectious diseases. Although many reviews on A3s in the virology field, there is no (or few) review(s) summarizing the A3 genetic variations and viral infections in detail, as far as I know. The authors are well familiar with A3s as well as genome biology or bioinformatic analysis, and the information in this review is comprehensive and reliable. I have several major and minor comments.
Major:
- The authors should add a table summarizing the associations between representative SNPs and phenotypes in viral infectious disease.
- The authors may want to add a figure illustrating the A3H story since this story is interesting but complicated. In the figure, it would be better to summarize 1) the haplotypes and their amino acid difference, 2) the relationship between haplotypes and the stable/unstable A3H classification, 3) the relationship between stable/unstable A3H and hyper/hypo Vif, and 4) the geographic distribution/association of stable/unstable A3H and hyper/hypo Vif. In addition, the author may want to include the story about splicing and Hap II SV200 in this figure.
Minor:
- If there is a systematic review on A3 variations and viral infectious diseases, please refer to it.
- Are the associations summarized in this manuscript statistically significant? Did the authors of the original articles perform multiple test corrections?
- Regarding Fig. 1: The font size of “RT blockage” at the upper right corner is too small.
- Regarding Fig. 2: Please add “and in-frame fusion” after “~30 kb deletion”.
- Line 94: Why “Importantly”? Please explain the reason.
- Line 150: In “found seven SNPs”; please add the description about these SNPs.
- Line 226: Why there are many variations in A3H compared to the other A3 genes? If the authors had any insights from the evolutionary aspect, please share them.
- Line 241: Please add a description of the relationship between these amino acid substitutions and haplotypes.
- Line 294: “Open questions (or future direction) and conclusion”.
- Line 318–321 (or other places): It would be better to add a description that the rate of stop codon emergence would be different between A3s having GG > GA- and GA > AA-type preferences.
Author Response
REVIEWER 3:
Comments and Suggestions for Authors
In this review article, the authors, Sadeghpour et al., summarize the association between human APOBEC3 (A3) variations and phenotypes in viral infectious diseases. Although many reviews on A3s in the virology field, there is no (or few) review(s) summarizing the A3 genetic variations and viral infections in detail, as far as I know. The authors are well familiar with A3s as well as genome biology or bioinformatic analysis, and the information in this review is comprehensive and reliable. I have several major and minor comments.
Major:
The authors should add a table summarizing the associations between representative SNPs and phenotypes in viral infectious disease.
Response: Table 1 has been added.
The authors may want to add a figure illustrating the A3H story since this story is interesting but complicated. In the figure, it would be better to summarize 1) the haplotypes and their amino acid difference, 2) the relationship between haplotypes and the stable/unstable A3H classification, 3) the relationship between stable/unstable A3H and hyper/hypo Vif, and 4) the geographic distribution/association of stable/unstable A3H and hyper/hypo Vif. In addition, the author may want to include the story about splicing and Hap II SV200 in this figure.
Response: We have added Fig 3.
Minor:
If there is a systematic review on A3 variations and viral infectious diseases, please refer to it.
Response: To the best of our knowledge we have included all published papers/reviews on A3 variations.
Are the associations summarized in this manuscript statistically significant? Did the authors of the original articles perform multiple test corrections?
Response: We thank the reviewer for this question. Most studies report the usage of appropriate statistical analysis, but it is not trivial to know the details of the analysis.
Regarding Fig. 1: The font size of “RT blockage” at the upper right corner is too small.
Response: Fixed.
Regarding Fig. 2: Please add “and in-frame fusion” after “~30 kb deletion”.
Response: Done.
Line 94: Why “Importantly”? Please explain the reason.
Response: The word importantly is removed and the sentence was re-written as “The variant 188I which has a higher enzymatic activity, is frequent in sub-Saharan African populations”.
Line 150: In “found seven SNPs”; please add the description about these SNPs.
Response: The description comes immediately after this sentence. Please see lines 696-700. “Three of these SNPs are upstream of APOBEC3G with potential regulatory roles [-571G/C (rs5757463), -199G/A (rs34550797), and -90C/G (rs5750743)], two are within introns [197193T/C (rs3736685) and 199376G/C (rs2294367)] and two within exons [F119F (rs5757465) and H186R (rs8177832)].”
Line 226: Why there are many variations in A3H compared to the other A3 genes? If the authors had any insights from the evolutionary aspect, please share them.
Response: This is an interesting question for which there is no answer yet.
Line 241: Please add a description of the relationship between these amino acid substitutions and haplotypes.
Response: The ‘rs’ numbers were added and its stated that they are genetically linked to haplotypes (Lines 1006-1008).
Line 294: “Open questions (or future direction) and conclusion”.
Response: It was changed to ‘Future directions and conclusion.
Line 318–321 (or other places): It would be better to add a description that the rate of stop codon emergence would be different between A3s having GG > GA- and GA > AA-type preferences.
Response: The following text was added to lines 1451-1453.
“This is particularly true for stop-codons, which are generated at a higher rate by APOBEC3G (GG>AG) compared to other APOBEC3 enzymes (GA>AA).”
Reviewer 4 Report
Major Comments:
Here the authors provide a comprehensive review of how genetic variation as well as differential splicing contributes to varied HIV restriction phenotypes and population-level associations. A large amount of data describing the association, or lack thereof, between APOBEC3 polymorphisms and HIV infection is catalogued. Overall, this review summarizes a large amount of data, a need not currently met in the field. The critiques provided here are mostly minor, although I do feel that in many cases too many details are included that are not relevant making the review difficult to read at times. The review would benefit from an attempt to make the key points of each study described more concise. Further, too much weight is given to variation having functional consequences or being due to selective forces. The possibility of the lack of selection, stochastic variation and linkage should be given more consideration and discussion. The presence of variation does not necessarily imply selection; this is overstated throughout the review.
Figure 1 – more details should be added to describe what is shown. For example, why is Vif shown binding to A3D and A3F? As this review does not cover APOBEC3 mechanism this figure could be omitted or modified to demonstrate how variation in APOBEC3s impact molecular mechanism?
Can the SNPs and associations or lack-thereof in the various studies be summarized in a table? This would add a significant amount of clarity.
Minor Comments:
An example of irrelevant details being included is: in Lines 50-51 the authors state that “the APOBEC3A/3B deletion is in high linkage disequilibrium with rs12628403”. Why is this relevant?
Lines 50-54: the important point here is not that the APOBEC3A coding sequence is maintained but rather that there is a functional APOBEC3B allele deletion as discussed in Kidd et al.
Lines 13 and 33: the expansion of the APOBEC3 locus is not only in humans but conserved across primates
Line 44: what is meant by ‘at a genomic level’? the changes in transcripts described by the authors are impacted by a genomic mutation. And all of these variations play out at the protein level (what do the authors mean by ‘proteomic’?). It’s not clear what the authors are trying to say here.
Lines 83 – 85: it could be prudent for the authors to provide one or more ideas as to why this is the case to allow the reader to better understand the discordant results.
Line 275-277: The first four haplotypes of APOBEC3H were first described here: https://pubmed.ncbi.nlm.nih.gov/18779051/. Harari et al did identify alternative splicing of APOBEC3H for the first time.
Line 290-291: this is entirely unique to sub-Saharan African populations or just more prevalent?
Line 92: APOBECH (typo)
Line 142: needs a reference
Line 239: “Gourraud et al. genotypes”?
Line 275: “considerable expression” should be defined
Lines 275-276: these are splice variants, not haplotypes
Line 290: what is “exogenizing”?
Author Response
REVIEWER 4:
Comments and Suggestions for Authors
Major Comments:
Here the authors provide a comprehensive review of how genetic variation as well as differential splicing contributes to varied HIV restriction phenotypes and population-level associations. A large amount of data describing the association, or lack thereof, between APOBEC3 polymorphisms and HIV infection is catalogued. Overall, this review summarizes a large amount of data, a need not currently met in the field. The critiques provided here are mostly minor, although I do feel that in many cases too many details are included that are not relevant making the review difficult to read at times. The review would benefit from an attempt to make the key points of each study described more concise. Further, too much weight is given to variation having functional consequences or being due to selective forces. The possibility of the lack of selection, stochastic variation and linkage should be given more consideration and discussion. The presence of variation does not necessarily imply selection; this is overstated throughout the review.
Response: We removed some of the details and re-wrote the introduction to help with the jargons used in the A3 sections. We deliberately included the details to enable the reader judge the validity of the reported results based on the number of samples, polulations, etc.
This review attempts to cover all of the SNPs but is focused on the functionally relevant variations. Please see line 20-21.
Figure 1 – more details should be added to describe what is shown. For example, why is Vif shown binding to A3D and A3F? As this review does not cover APOBEC3 mechanism this figure could be omitted or modified to demonstrate how variation in APOBEC3s impact molecular mechanism?
Response: We revised Fig 1 and now show the degradation for A3D and A3F as well. We also re-wrote the introduction based on comments from the other reviewers who asked for details and molecular mechanisms.
Can the SNPs and associations or lack-thereof in the various studies be summarized in a table? This would add a significant amount of clarity.
Response: Table 1 was added.
Minor Comments:
An example of irrelevant details being included is: in Lines 50-51 the authors state that “the APOBEC3A/3B deletion is in high linkage disequilibrium with rs12628403”. Why is this relevant?
Response: A3B deletion does not have an ‘rs’ number. rs12628403 is used as its surrogate. A clarification has now been added to the text (Line 229)
Lines 50-54: the important point here is not that the APOBEC3A coding sequence is maintained but rather that there is a functional APOBEC3B allele deletion as discussed in Kidd et al.
Response: We revised the text to “Genomes with this deletion do not express APOBEC3B mRNA but instead they express a fused APOBEC3A/B mRNA, which codes for a protein identical to APOBEC3A.” See lines 230-231.
Lines 13 and 33: the expansion of the APOBEC3 locus is not only in humans but conserved across primates
Response: We changed “human” to “primates”.
Line 44: what is meant by ‘at a genomic level’? the changes in transcripts described by the authors are impacted by a genomic mutation. And all of these variations play out at the protein level (what do the authors mean by ‘proteomic’?). It’s not clear what the authors are trying to say here.
Response: We changed the text to “APOBEC3 genes have many coding and noncoding variations. They are also diverse in terms of the transcripts (splice variants) they produce. The majority of variations reported for APOBEC3 enzymes are at a DNA level, particularly single nucleotide polymorphisms (SNPs). Few studies have also investigated APOBEC3 mRNA splicing.” Lines 214-217
Lines 83 – 85: it could be prudent for the authors to provide one or more ideas as to why this is the case to allow the reader to better understand the discordant results.
Response: We currently do not have an explanation or hypothesis. We added the following to the text. “But given that APOBEC3B localizes to the nucleus and is not expressed in CD4 T cells, this enzyme is expected to have little/no impact on the HIV-1 disease status. The reason for the observed associations is not clear.” Lines 324-327.
Line 275-277: The first four haplotypes of APOBEC3H were first described here: https://pubmed.ncbi.nlm.nih.gov/18779051/. Harari et al did identify alternative splicing of APOBEC3H for the first time.
Response: We revised the text to: ‘Harari et al. analyzed PBMCs from 12 healthy donors and showed that all of APOBEC3H haplotypes 1-IV were resistant to Vif.’ Line 1070.
Line 290-291: this is entirely unique to sub-Saharan African populations or just more prevalent?
Response: We revised the text to: ‘The CTC deletion is almost exclusively embedded in the HapII genome, which is present (at least one copy) in ~80% of sub-Saharan African populations. The CTC deletion induces the inclusion a fragment of an intronic antisense L1 element as a new exon (exon 4b) in APOBEC3H HapII mRNA. The result of this L1 exonization is a new splice variant (SV200) with an extended and different C-terminus.’ Lines 1389-1395.
Line 92: APOBECH (typo)
Response: Fixed.
Line 142: needs a reference
Response: Done.
Line 239: “Gourraud et al. genotypes”?
Response: Fixed.
Line 275: “considerable expression” should be defined
Response: “considerable expression” was removed.
Lines 275-276: these are splice variants, not haplotypes
Response: We revised the text to: ‘Harari et al. analyzed PBMCs from 12 healthy donors and showed that all of APOBEC3H haplotypes 1-IV were resistant to Vif.’ Line 1070
Line 290: what is “exogenizing”?
Exonization refers to the recruitment of an exon from regions annotated as non-coding (see https://pubmed.ncbi.nlm.nih.gov/21787833/). To clarify, we revised the text to “The CTC deletion induces the inclusion a fragment of an intronic antisense L1 element as a new exon (exon 4b) in APOBEC3H HapII mRNA. The result of this L1 exonization is a new splice variant (SV200) with an extended and different C-terminus.” Lines 1391-1394
Round 2
Reviewer 1 Report
Overall, Table 1 improves the manuscript substantially. I have a few suggestions for further improvements in the paper.
- Lines 33 and 44- The words cytidine and uridine are used incorrectly. Both are ribonucleosides, not deoxyribonucleosides. What deaminates is the base cytosine to uracil. This applies to both RNA and DNA.
- Line 61- change "excusably" to "exclusively".
- Lines 71 through 73- This is old and incorrect information. APOBEC3 enzymes do not edit genomes of herpes viruses. See PMID 30420783, 31493648 and 31534038.
- The Introduction should end by stating that this review will focus mainly on HIV and HBV viruses.
- Line 110- Missing recent work as stated in point #3.
- Line 143- Change "Contrary to" to "In contrast with".
- Line 186- Meaning of "APOBEC3D is not a diverse protein" is unclear.
- Table 1. Change the title of the column "Studied populations" to "Studied populations and reference".
Author Response
We thank the Reviewer again for his/her comments. All new changes are highlighted in yellow in the revised manuscript.
- Lines 33 and 44- The words cytidine and uridine are used incorrectly. Both are ribonucleosides, not deoxyribonucleosides. What deaminates is the base cytosine to uracil. This applies to bothRNA and DNA.
Response: Fixed
- Line 61- change "excusably" to "exclusively".
Response: Fixed
- Lines 71 through 73- This is old and incorrect information. APOBEC3 enzymes do not edit genomes of herpes viruses. See PMID 30420783, 31493648 and 31534038.
Response: We totally agree with the Reviewer that APOBEC3 enzymes do not play a role in hypermutating herpesviruses, at least in vivo. This is based on our recent analysis of all reported herpesvirus sequences which returned no evidence of hypermutation. However, an older study by the Wain-Hobson lab (PMID: 31493648) and also a recent study by the Harris lab (PMID: 30420783) reported herpesviruses hypermutation in vitro (See Fig. 4 in PMID: 30420783). To clarify these, we revised the text to “The function of APOBEC3 enzymes is not limited to HIV-1 restriction. These enzymes have also been implicated in the inhibition and/or evolution of several other viruses [51] such as hepatitis B virus (HBV) [52-55], polyomaviruses (e.g. JC and BK) [56, 57], human T cell leukemia virus-1 (HTLV-1) [58, 59], human papillomavirus (HPV) [60-62], and herpesviruses [63, 64] such as Epstein-Barr virus (EBV) [65, 66], herpes simplex virus-1 (HSV-1) [65] and Kaposi’s sarcoma-associated herpesvirus (KSHV) [67]. It is important to note that data on the inhibition/hypermutation of herpesviruses by APOBEC3 enzymes are not consistent [65, 66, 68, 69]. Overall, unlike HIV-1, evidence for in vivo hypermutation of these viruses is extremely limited. Therefore, deaminase-independent mechanisms are likely responsible for the inhibition of these viruses by APOBEC3 enzymes.”
All references suggested by the Reviewer have been included (See Refs 65, 66, 68, 69).
- The Introduction should end by stating that this review will focus mainly on HIV and HBV viruses.
Response: We thank the Reviewer for this comment and agree that most of the studies are related to HIV and HBV. But this review does not focus specifically on a given virus and covers all reported reported viruses.
- Line 110- Missing recent work as stated in point #3.
Response – Please see response to point #3 above.
- Line 143- Change "Contrary to" to "In contrast with".
Response – Fixed.
- Line 186- Meaning of "APOBEC3D is not a diverse protein" is unclear.
Response: The text has been revised to “There are not many coding variations within APOBEC3D, therefore studies focusing on APOBEC3D variations are limited”.
- Table 1. Change the title of the column "Studied populations" to "Studied populations and reference".
Response – Fixed.